# Microbiome in Male Genital Mucosa (Prepuce, Glans, and Coronal Sulcus): A Systematic Review

**DOI:** 10.3390/microorganisms10122312

**Published:** 2022-11-22

**Authors:** Micael F. M. Gonçalves, Ângela Rita Fernandes, Acácio Gonçalves Rodrigues, Carmen Lisboa

**Affiliations:** 1Division of Microbiology, Department of Pathology, Faculty of Medicine, University of Porto, 4200-319 Porto, Portugal; 2CINTESIS/RISE, Center for Health Technology and Services Research/Rede de Investigação em Saúde, Faculty of Medicine, University of Porto, 4200-319 Porto, Portugal; 3Department of Dermatology and Venereology, Centro Hospitalar Universitário São João, 4200-319 Porto, Portugal

**Keywords:** male genital mucosa microbiome, penis microbiome, reproductive system, urogenital system, genital disease

## Abstract

The human body represents a complex and diverse reservoir of microorganisms. Although the human microbiome remains poorly characterized and understood, it should not be underestimated, since recent studies have highlighted its importance in health. This is especially evident when considering microbiota in the male reproductive system, responsible for men’s fertility and sexual behavior. Therefore, the aim of this systematic review is to provide an overview of the microbial communities of the healthy male genital mucosa and its role in disease. This study was performed according to the Preferred Reporting Items for Systematic Reviews and Meta-Analyses (PRISMA) guidelines. The search was limited to the English language and studies published until August 2022 that included culture-independent techniques for microbiome characterization in male genital mucosa. Ten articles were included. The bacterial composition of the male genital mucosa consists of several genera including *Prevotella*, *Finegoldia, Peptoniphilus*, *Staphylococcus, Corynebacterium*, and *Anaerococcus*, suggesting that the male genital microbiome composition shows similarities with the adjacent anatomical sites and is related with sexual intercourse. Moreover, male circumcision appears to influence the penile microbiome. Despite the lack of knowledge on the male genital mucosa microbiome in disease, it was reported that *Staphylococcus warneri* and *Prevotella bivia* were associated with balanoposthitis, whereas *Enterobacteriaceae*, *Prevotella,* and *Fusobacterium* were more abundant in male genital lichen sclerosus. The limited data and paucity of prospective controlled studies highlight the need for additional studies and established criteria for sampling methods and the microbiome assay procedure. Such a consensus would foster the knowledge about the composition of the genital microbiome of healthy males and its role in disease.

## 1. Introduction

The complex and diverse human body microbiome has an enormous potential to influence human pathophysiology. Studies of the human microbiome suggest that several microbial communities play important roles in host homeostasis, regulating health, infection risk, and disease progression [1].

Recent advances in high throughput sequencing technologies, such as next-generation sequencing (NGS) techniques, the whole genome shotgun, and metagenomic sequencing, have been stimulating a new wave of research on the human microbiome [2]. These sequencing analyses allow for a more accurate detection of resident microbial DNA, revealing a considerable diversity of hidden microbes on most exposed surfaces of the human body, including the reproductive system [3].

The microbiome of male genital mucosa is extensively less studied compared to other areas of microbiome research, such as inflammatory diseases [4], skin conditions [5], gut and colorectal cancer [6], prostate cancer [7], and even the female reproductive tract [8]. Nevertheless, emerging evidence indicates that the male genital tract microbiome is a field of increasing scientific interest since it has important implications for the male reproductive health, men’s fertility, and sexual behavior [9,10,11]. Attention is now focused on understanding the alterations in the microbiome in case of dysbiosis [3].

Similar to vaginal mucosa, male genital mucosa is the most significant and adaptable structure in the male reproductive system, capable of mounting a full range of immune responses [12]. Thus, elucidation of the composition of the healthy male genital mucosa microbiome and its variations is of paramount importance. Although some studies have already investigated the role of the male reproductive tract microbiome [3], they are of limited scope and mainly focused on the urine or the coronal sulcus. The lack of comprehensive studies addressing the male genital mucosa microbiome undermines the prognostic determinants that include risk factors such as bacterial and virus infections. Thus, the aim of this systematic review is to identify and summarize studies focused on the microbial community’s composition in the different anatomical parts of the healthy and diseased male genital mucosa.

## 2. Materials and Methods

This systematic review was executed following the Preferred Reporting Items for Systematic Reviews and Meta-Analyses (PRISMA) guidelines [13]. The methods used, including title and abstract screening, full text review, inclusion, and data extraction criteria, were approved by all the authors.

### 2.1. Search Strategy

We performed a search in MEDLINE (PubMed), Web of Science, Scopus, and Google Scholar on 2 September 2022. The search strategy included keywords related to the scientific literature concerning the microbiome of the male genital mucosa, such as (alone or in combination): microbiome, male, genital tract, genital mucosa, penis, glans, balanopreputial sulcus, coronal sulcus, prepuce, and foreskin. Appendix A comprises the complete search strategies for each database.

### 2.2. Inclusion and Exclusion Criteria

Articles and clinical reports published until August 2022 and written in English were eligible for inclusion. Eligible studies had to be related to microbiome research on the male genital mucosa and include culture-independent techniques for the whole spectrum of detected microbiota.

Reviews, books and book chapters, poster presentations, theses and dissertations, and articles of which only the abstract was available, studies with only culture-dependent techniques, animal studies, and articles written in languages other than English were excluded.

### 2.3. Study Selection

Titles and/or abstracts of studies retrieved using the above-mentioned search strategy were screened independently by two authors (M.F.M.G. and Â.R.F.) to identify studies eligible for full-text screening. The full texts of these eligible articles were retrieved and independently assessed by the same two authors. Any disagreements concerning the eligibility of studies were resolved through discussion with a third author (C.L.).

### 2.4. Data Extraction

Data were extracted from the included studies for an assessment of study quality. Predefined extracted information included: field of study; study population; participant demographics and baseline characteristics (including age); study setting (including ethnicity and country); target anatomical organ; study methodology (including sample type method and type of microbiome analysis); main outcomes and limitations.

### 2.5. Study Quality Assessment

Two authors (M.F.M.G. and Â.R.F.) independently assessed the quality evaluation of individual studies according to the National Institutes of Health (NIH) Study Quality Assessment Tools for observational studies [14] and Cochrane risk-of-bias tool for randomized trials [15]. Disagreements resulting from this process were resolved through discussion with the other authors (A.G.R. and C.L.).

## 3. Results

### 3.1. Number of Retrieved Papers

A flowchart of the search strategy and study selection process of the articles is shown in Figure 1. The search yielded 149 articles, which resulted in 144 unique articles after removing duplicates. After screening the titles and abstracts, 129 articles were excluded based on title and study type. After reading the full-text articles, 15 articles were eligible for inclusion, and 5 were excluded (non-study target area, *n* = 2; no microbiome, *n* = 2; articles written in languages other than English, *n* = 1). Table 1 provides an overview of the 10 final selected articles and summarizes the characteristics and reported taxonomic findings.

From the 10 studies that were included in this review, all of them employed 16S rRNA gene amplicon sequencing for analysis of the male genital mucosa microbiome. However, the amplification regions of the 16S rRNA genes were different (V3–V4, *n* = 4; V4, *n* = 3; V3–V6, *n* = 1; V4–V6, *n* = 1; and combination of V1–V3, V3–V5 and V6–V9, *n* = 1). None of these studies analyzed the fungal fraction of the microbiota through sequencing of the internal transcribed spacer (ITS) region of the rDNA. In total, the male genital mucosa microbiome analysis has been performed on 697 men (Table 1).

### 3.2. Pediatric Foreskin Microbiome

The study by Storm et al. [23] is the only one included in this review that performs the analysis of the foreskin microbiome at the pediatric age. Forty-eight males and twenty-six females less than 18 years old without previous antibiotic exposure were recruited to study the urobiome. In contrast to the female urobiome (urethra, perineum, and vagina), no significant changes were observed in the urethral and foreskin microbiome composition over time in males. Only the perineal microbiome differed significantly between prepubertal/toilet-trained (3–12 years) and post-pubertal (>12 years) males. The foreskin microbiome showed a higher abundance of *Prevotella*, *Staphylococcus*, *Corynebacterium*, *Peptoniphilus*, *Mobiluncus*, and *Winkia*. In the perineum of pre-toilet-trained boys, *Veillonella*, *Bifidobacterium*, and *Enterococcus* genera were abundant. In prepubertal boys, *Peptoniphilus* sp., *Anaerococcus* sp., *Faecalibacterium* sp., and *Finegoldia* sp. were the most abundant organisms. In post-pubertal boys, *Corynebacterium* spp. was abundant; and *C. tuberculosteraricum* and *S. aureus* were the most common species. This study demonstrated that a core group of urinary bacteria is present in early infancy and changes throughout childhood, with differences between males and females.

### 3.3. Circumcision and Coronal Sulcus Microbiome

Three studies investigated the effect of the male circumcision on the coronal sulcus microbiome of healthy men [18,20,22]. Nelson et al. [20] compared the microbiome of the coronal sulcus and urine from six uncircumcised and twelve circumcised adolescent men between 14 and 17 years old. Differences in the coronal sulcus and urine microbiome composition were found, the coronal sulcus microbiome being more stable and strongly influenced by circumcision. The coronal sulcus contained high proportions of the genera *Corynebacteria*, *Staphylococcus*, *Anaerococcus*, *Peptoniphilus*, *Prevotella*, *Finegoldia*, and *Porphyromonas*, while in urine, *Streptococcus*, *Lactobacillus*, *Gardnerella*, and *Veillonella* were the most predominant genera. Some taxa including *Prevotella* and *Porphyromonas* were more abundant in uncircumcised men but were not the main components of the coronal sulcus microbiome. The authors also detected bacterial vaginosis associated taxa, including *Atopobium* sp., *Megasphaera* sp., *Mobiluncus* sp., *Prevotella* sp., and *Gemella* sp., in the coronal sulcus from sexually experienced and inexperienced men.

Price et al. [22] characterized the coronal sulcus microbiome before and after circumcision from twelve HIV-negative participants aged 15–49 years during a randomized trial of male circumcision for HIV prevention. Price and colleagues reported that the coronal sulcus microbiome before circumcision was more heterogeneous than after circumcision and similar to several core community types observed in the vagina. The most prevalent genera in the coronal sulcus before circumcision were *Anaerococcus*, *Peptoniphilus*, *Finegoldia*, and *Prevotella*. After circumcision, *Staphylococcus* and *Corynebacterium* were the most prevalent genera. In addition, the authors observed a decrease in anaerobic bacteria due to the elimination of anoxic microenvironments under the foreskin. They concluded that circumcision alters the coronal sulcus microbiome, and anaerobic and vaginosis associated bacteria are the most abundant before circumcision due to the existence of the moist anoxic microenvironment of the subpreputial space. Additionally, Liu et al. [18] also evaluated the coronal sulcus microbiome of seventy-nine HIV-negative men (15–49 years) randomly assigned to receive male circumcision and seventy-seven men that remained uncircumcised. Significant changes were observed in the coronal sulcus bacterial load, i.e., male circumcision was associated with a decrease in the coronal sulcus bacterial load relative to uncircumcised men. The prevalence and absolute abundance of 15 coronal sulcus bacteria, among which 12 were anaerobes, decreased significantly in the circumcised men, whereas aerobes increased after circumcision. This reduction includes species of *Porphymonas*, *Prevotella*, *Negativicoccus*, *Dialister*, *Mobiluncus*, and six genera from *Clostridiales* family XI. No significant decrease was observed in other anaerobe species such as *Atopobium* sp., *Sneathia* sp., and *Megasphaera* sp. after circumcision. Seven coronal sulcus bacteria were found to become more prevalent after circumcision; e.g., the aerobic *Kocuria* spp. and the facultative anaerobic *Facklamia* spp. were found to be more prevalent exclusively among circumcised males. The authors concluded that the bacterial changes identified in circumcised and uncircumcised HIV-negative men may play an important role in the HIV risk reduction conferred by male circumcision.

To test the hypothesis that penile anaerobe abundance directly promotes HIV risk by inducing a proinflammatory response in the foreskin, Liu et al. [19] used a case-control study. The authors compared the microbiome and cytokine levels in the penile coronal sulcus in 46 uncircumcised men who seroconverted and 136 uncircumcised men who remained HIV seronegative (controls) during a randomized-controlled trial of medical male circumcision in Rakai, Uganda. Total penile bacterial loads were similar in males HIV-positive and HIV-negative. However, males who acquired HIV had significantly higher abundances of penile anaerobes than men who remained HIV negative during the trial, including *Prevotella*, *Dialister*, *Mobiluncus*, *Murdochiella*, and *Peptostreptococcus* genera. Without any adjustment for risk factors, the authors found a consistent relationship between anaerobe abundance and HIV seroconversion among 5 of the 10 anaerobic genera. After adjustment, the association between the abundance of anaerobic bacteria and the odds of HIV seroconversion strengthened. Species associated with the greatest increased risk of HIV seroconversion were *Prevotella*, followed by *Dialister*, and six other genera of anaerobic bacteria (*Peptoniphilus*, *Finegoldia*, *Porphyromonas*, *Mobiluncus*, *Peptostreptococcus*, and *Murdochiella*). The authors concluded that uncircumcised men who became HIV infected during a 2-year clinical trial had higher levels of penile anaerobes in comparison with uncircumcised men who remained HIV negative.

### 3.4. Male Genital Mucosa Microbiome and Bacterial Vaginosis

Two studies investigated the role of the microbiome of male genital mucosa associated with bacterial vaginosis. Zozaya et al. [25] examined the microbiome composition of genital bacteria in monogamous couples, including penile skin (glans, the coronal sulcus, and the shaft of the penis) and urethral specimens from predominantly African American men whose sexual partners were diagnosed with bacterial vaginosis, as defined by the Nugent and Amsel score [26]. Penile skin diversity of sixty-five males (23 circumcised and 42 uncircumcised) in the bacterial vaginosis-couples group and sixty-five control males (partners of women without bacterial vaginosis) (35 circumcised and 30 uncircumcised) were analyzed. The authors observed that the penile skin diversity of male partners of women with bacterial vaginosis was significantly higher than that of control males, but urethral diversity did not differ between groups. Moreover, the results showed that in bacterial vaginosis-couples, the penile skin communities were significantly more similar to the vaginal communities of their sexual partner. The authors concluded that sexual transmission of bacterial vaginosis associated bacteria is a common occurrence during sex. No clear separation was found between the penile skin microbiome of circumcised and uncircumcised men, for both males’ partners groups. In general, the penile skin microbiome in this study showed a higher abundance of *Peptoniphilus* sp., *Anaerococcus* sp., *Pv. 123-f-82*, and *Lactobacillus iners* in both males’ groups.

Plummer et al. [21] observed that concurrent partner treatment for bacterial vaginosis significantly altered the composition of the genital microbiome of both partners in 27 couples. After 12 weeks of antibiotic administration, a reduction in bacterial vaginosis associated bacteria was observed. The abundance of 11 taxa, including the genera *Anaerococcus*, *Finegoldia*, *Peptoniphilus*, *Prevotela*, and *Dialister*, reduced significantly, while *Staphyloccocus* sp. significantly increased during treatment. The authors concluded that prescription of antibiotic treatment for male partners may be a strategy to achieve a sustained bacterial vaginosis cure to improve reproductive and sexual health for women.

### 3.5. The Microbiome in Genital Mucosa Inflammation

Two recent studies investigated the role of the microbiome in balanoposthitis and lichen sclerosus inflammation that affect both the glans penis and prepuce, comparing patients to controls. The study conducted by Li et al. [17] found no significant differences in the genital mucosa microbiome between 26 men with balanoposthitis and 29 healthy men. However, differences were observed between men with balanoposthitis and healthy men with redundant prepuce. The dominant species in men with balanoposthitis were *Staphylococcus warneri* and *Prevotella bivia*, which were positively correlated with disease severity. In healthy men with redundant prepuce, the most prevalent species were *Ezakiella* sp. and *Porphyromonas somerae*.

Watchorn et al. [24] observed differences in the bacterial composition in the balanopreputial sac in 20 uncircumcised men aged 26–73 years with male genital lichen sclerosus (MGLSc) compared to 20 healthy uncircumcised men aged 19–63 years. *Enterobacteriaceae*, *Prevotella,* and *Fusobacterium* were identified as the most abundant taxa in the MGLSc group, but not in healthy men. In the control group, the most dominant genera were *Finegoldia*, *Staphyloccocus*, and *Corynebacterium*. The relative abundance of *Finegoldia* was lower in the MGLSc group than in the control group, and *Fusobacterium* was higher in the MGLSc group than in the control group.

### 3.6. Influence of Probiotic Supplementation on Glans Microbiome

A study carried out by Iniesta et al. [16] evaluated the effect of *Ligilactobacillus salivarius* PS116610 on the microbial composition of the urogenital tract in seventeen Caucasian (Spanish) infertile couples aged 20–40 years and under assisted reproduction treatment diagnosed with bacterial dysbiosis. Glans swabs from enrolled males were analyzed at the beginning of the study and after 3 and 6 months of treatment. Unlike the vagina microbiome, no significant changes were observed in the microbiome composition before and after the probiotic treatment in males, although a slight decrease in urogenital pathogens was registered. The glans microbiome from enrolled males is mainly composed by *Peptoniphilus* sp., *Finegoldia* sp., *Corynebacterium* sp., and *Staphylococcus* sp. The authors concluded that probiotic supplementation with *L. salivarius* PS116610 in couples with idiopathic infertility under assisted reproduction treatment improved the urogenital tract microbiome, solving the dysbiosis in 88.9% of the couples.

## 4. Discussion

To our knowledge, this is the first study that assesses the microbiome of healthy and diseased male genital mucosa. Due to the advances in high-throughput sequencing technologies, bioinformatic tools, and mass spectrometry techniques, omics approaches have enhanced our ability to characterize the diversity and function of microbiome. Some of the meta-omics technologies (metagenomics, metatranscriptomics, metaproteomics, and metametabolomics) have been used for clinical diagnosis in various diseases [27,28,29,30,31]. The 16S rRNA gene has been most frequently targeted due to its presence in all prokaryotes. However, to characterize the full genetic content of a community, metagenomic studies go beyond the 16S rRNA gene [27]. Here, all studies performed 16S rRNA sequencing, but focusing on different male age groups, diseases, and conditions. Studies ranged from childhood to early elderly, male genital inflammation (balanoposthitis and male genital lichen sclerosus), female genital inflammation (bacterial vaginosis), male circumcision, and HIV.

One of the main findings of this study is that there is very limited knowledge on the male genital mucosa (prepuce, glans, and coronal sulcus) microbiome. Several bacterial taxa have been indicated as resident microbiota on male genital mucosa such as *Prevotela*, *Finegoldia, Peptoniphilus*, *Staphylococcus, Corynebacterium*, and *Anaerococcus*. The higher prevalence of such taxa suggests its possible provenance from other sites, such as superficial and sebaceous skin, the inguinal region, the gut, or even vaginal associated taxa [8,32]. The genus *Prevotella* is the only strict anaerobe, frequently found in the gut [33], and it is negatively associated with sperm mobility [34]. Species of *Finegoldia*, *Peptoniphilus*, and *Anaerococcus* have been reported as colonizers of the skin and mucosal surfaces, such as the mouth, upper respiratory tract, gastrointestinal tract, and female genitourinary tract [35]. Notably, in the healthy male genital mucosa, these species were also found. Furthermore, coryneform bacteria were also identified in many studies on the male urogenital tract [36]. Frequently, these bacteria tend to be overlooked as commensals, but some authors associated these microorganisms with infection [17,24,37].

The representation of the prepuce, glans, and coronal sulcus microbiome provided in Figure 2 suggests that different bacteria can colonize male genital anatomic sites and determine site-specific microbiota. The microbiome of the glans and coronal sulcus appears to be similar in all studies included in this review. However, the only study available on the microbiome of the prepuce focused on subjects under 18 years old [23]. The authors stated that the most abundant microorganisms were the bacterial genera *Prevotella*, *Staphylococcus*, *Corynebacterium*, *Peptoniphilus*, *Mobiluncus*, and *Winkia*. We should highlight that the reporting of one study hampers a comprehensive comparison of the microbiome of the prepuce in older men. Moreover, this study analyzed a small sample size and did not report the pubertal status of an older cohort, which should be encouraged in future studies.

One of the key points of this review was to identify the microbial community’s composition in the different anatomical parts of the healthy and diseased male genital mucosa. Therefore, the current study confirmed the previous findings in which the male genital mucosa microbiome can be influenced by many factors, such as physical barriers, inflammation, infection, interaction within adjacent niches, antimicrobial peptides, and lipids [16,38,39]. Thus, changes in host-genital microbiome interactions may be linked to disease development [23].

Male circumcision is one of the most common surgical procedures in the world, and, in some situations, this type of surgery is necessary for medical reasons [40]. The four studies included in this review that investigated the effect of the male circumcision on the coronal sulcus microbiome share similar conclusions. Male circumcision appears to impact the coronal sulcus microbiome due to the elimination of the anoxic subpreputial microenvironment. Prepuce removal unbalances the anaerobic sites; consequently, the abundance of anaerobic bacteria decreases, exerting a selective pressure for aerobic bacteria, along with bacterial competition [18,22]. Moreover, higher levels of penile anaerobes detected in uncircumcised men were associated with higher production of immune factors that recruit HIV target cells [19] and the abundance of vaginal-associated taxa [20]. Moreover, the studies by Zozaya et al. [25] and Plummer et al. [21] showed that sexual transmission of vaginal-associated bacteria, such as bacterial vaginosis, is a common occurrence; organisms might be exchanged during sexual intercourse. In fact, some studies already showed that sexual history could be a determinant of the penis microbiome composition. Dong et al. [41] and Nelson et al. [42] showed that the penis and the urethra can be colonized by a variety of bacterial vaginosis associated taxa.

Comparing the present results of the male genital mucosa microbiome with the recent systematic review by Pagan et al. [8] of the vulvar microbiome, the taxa present on the vulva are equivalent to some of the most prevalent taxa of male genital mucosa, such as the genera *Corynebacterium*, *Staphylococcus*, and *Prevotella*, but with the exception of *Lactobacillus*. In fact, *Lactobacillus* are well-known lactic-acid-producing bacteria that colonize the female genital tract [43]. The authors suggest that these bacteria might be exhibited in vaginal, cutaneous, and intestinal niches. As mentioned above, the male genital mucosa microbiome can be influenced by adjacent or nearby niches, which may include the perineum, inguinal region, and intestine. Both male and female genital mucosa are characterized as a moist environment [8,18] creating the favorable conditions for certain microorganisms. Therefore, the understanding of shared genital microbiota between sexual partners might be a milestone in genitourinary diseases.

Nevertheless, several limitations of the current literature can be identified. Firstly, the sample population of practically all studies is rather low (ranging between 12 to 50 participants), with the exception of three (out of ten) studies that encompass more than 130 participants per study [18,19,25]. In addition, participants’ ethnicity data are most often absent. Three studies identified the ethnicity record of participants: Caucasian [16], African American [25], and one study included a diverse population, i.e., Mixed Black, Caucasian, and Latin [20]. Furthermore, most of these studies focused on a specific sample of individuals enrolled in medical clinics with different specialties, such as in sexually transmitted disease clinics, HIV prevention, or in dermatology clinics. Moreover, the current literature includes different male age groups. Only two studies addressed the analysis of the microbiome at the pediatric age [20,23] but with different purposes. Nelson et al. [20] evaluated the bacterial communities of the coronal sulcus and distal urethra of circumcised and uncircumcised adolescent males, and Storm et al. [23] studied the urethral and foreskin microbiome composition over time in males aged 0 to 18 years. Therefore, results from these studies should be carefully interpreted, and type I errors should be noted for future research.

Moreover, different sample collection methods and the possible risk of contamination during sampling prevented direct comparison between studies. It is worth noting that, when the catheterized urine technique is applied, there is a reduction in contamination by the urethra, avoiding the overrepresentation of microorganisms [23]. Genital hygiene and frequency of sexual intercourse should also be considered. Self-hygiene prior to sample collection can add bias to microbiome studies as well as self-collected samples, such as in the study by Plummer et al. [21] whose males self-collected a cutaneous penile swab.

Notably, none of the ten studies under the scope of this systematic review investigated the mycobiome, virome, and parasitome in male genital mucosa. All studies analyzed the bacteriome through 16S rRNA gene amplicon sequencing. The usefulness of the 16s rRNA gene sequencing technique is well known [44]. However, its range to the genus level is a limitation [21], and the choice of the hypervariable region target of the 16s rRNA gene can lead to a discrepancy in microbial quantification and influences the outcomes. As mentioned by Li et al. [17], the V4 region is considered a relatively low informative region for taxonomic assignment. Thus, choosing the ideal 16S rRNA hypervariable region will depend on the bacterial composition according to the studied environment [45]. For example, Kameoka et al. [46] showed that the V1–V2 region is more precise than V3–V4 for the gut microbiome of Japanese individuals, while Hoffman et al. [47] found that the V1–V3 and V2–V3 regions are preferred on the female urobiome. Additionally, Graspeuntner et al. [48] recognized that the V1–V3 region allows for a more complete assessment of the cutaneous and vaginal analysis. Furthermore, based on systematic comparisons of all nine 16S rRNA hypervariable regions, Heidrich et al. [45] concluded that V1–V2 is more suitable for male urinary microbiota profiling. To our knowledge and given the few available studies about the male genital mucosa microbiome, there is no information yet concerning the best 16S rRNA hypervariable region for microbial characterization of the prepuce, glans, and coronal sulcus. Interestingly, there was an enormous discrepancy in the 16S rRNA hypervariable region used in the studies included in this review. For example, to evaluate the glans and prepuce microbiome, Watchorn et al. [24] and Plummer et al. [21] used primers for the V3–V4 region, while Zozaya et al. [25] used the V4–V6 region. The same incongruity was observed for coronal sulcus microbiome studies, in which Liu et al. [19] used the V3–V4 region, and Liu et al. [18] used the V3–V6 region.

## 5. Conclusions

This systematic review represents the first report that encompasses the composition of microbial communities in the different anatomical parts of the healthy and diseased male genital mucosa. Following the analysis of the selected studies, we concluded that the knowledge about the microbiome of the male genital mucosa is still very limited. Besides that, some limitations can also be found across these studies, namely the small size of samples, the lack of ethnicity records, different methodological approaches (e.g., self-collected swab), and different microbiome analysis methods (e.g., 16S rRNA hypervariable regions). Therefore, these limitations can bias the results and comparisons between studies.

The microbiome composition of the male genital mucosa shows a high diversity of commensals organisms from the adjacent anatomical sites (perineum, skin, gut) or even urine. In addition, due to different types of sexual activities, the penis and urethra can be colonized by a variety of microorganisms, including bacterial vaginosis associated taxa as a result from partnered sexual activity. We cannot rule out that some of these microorganisms may play a specific role in the local milieu, providing the male genital mucosa a distinct signature that should be better elucidated in further studies. Acknowledging the microbiome of male genital mucosa is essential to understand and determine the interaction and role of the different microorganisms in male reproductive health, men’s fertility, and sexual behavior. Thus, changes in the microbiome composition may represent risk factors for microbial infection.

Future studies unraveling the mycobiome, virome, and parasitome on male genital mucosa are highly needed and should be addressed due to the prevalence of sexual transmitted fungal, viral and parasite infections, such as candidosis, trichomoniasis, HIV, and herpes. Moreover, studies comparing the sequencing performance of different hypervariable regions of 16S rRNA are necessary to identify which primer sets and combinations are more suitable for each anatomical part of the male genital mucosa. Despite the research motivation, a holistic and integrated approach combining other culture-independent techniques, such as pyrosequencing, whole metagenome sequencing and quantitative real-time PCR, and other meta-omics technologies are essential to ensure the accuracy of the results. This elucidates potential biomarkers intended for the diagnosis, prevention, and management of infections by screening for microbiological risk factors.

## Figures and Tables

**Figure 1 microorganisms-10-02312-f001:**
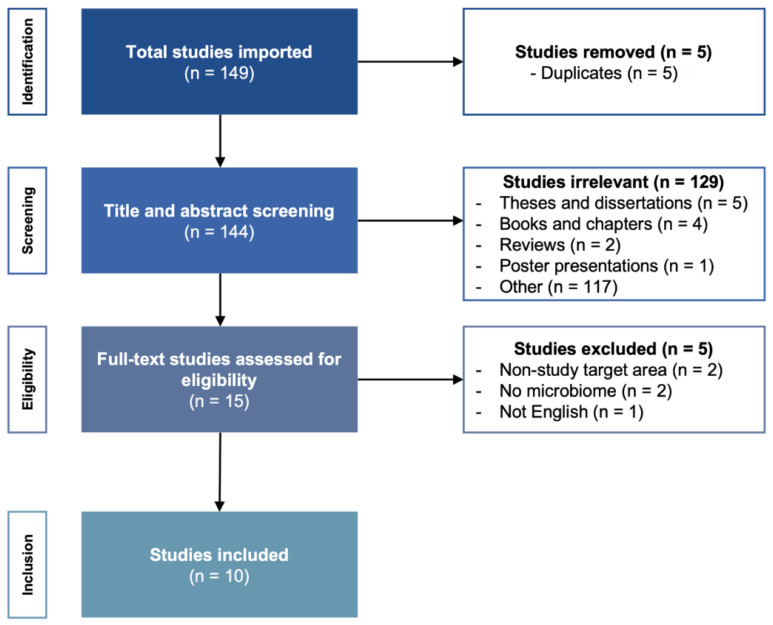
Flowchart of the search strategy for the selection of relevant articles following PRISMA methodology.

**Figure 2 microorganisms-10-02312-f002:**
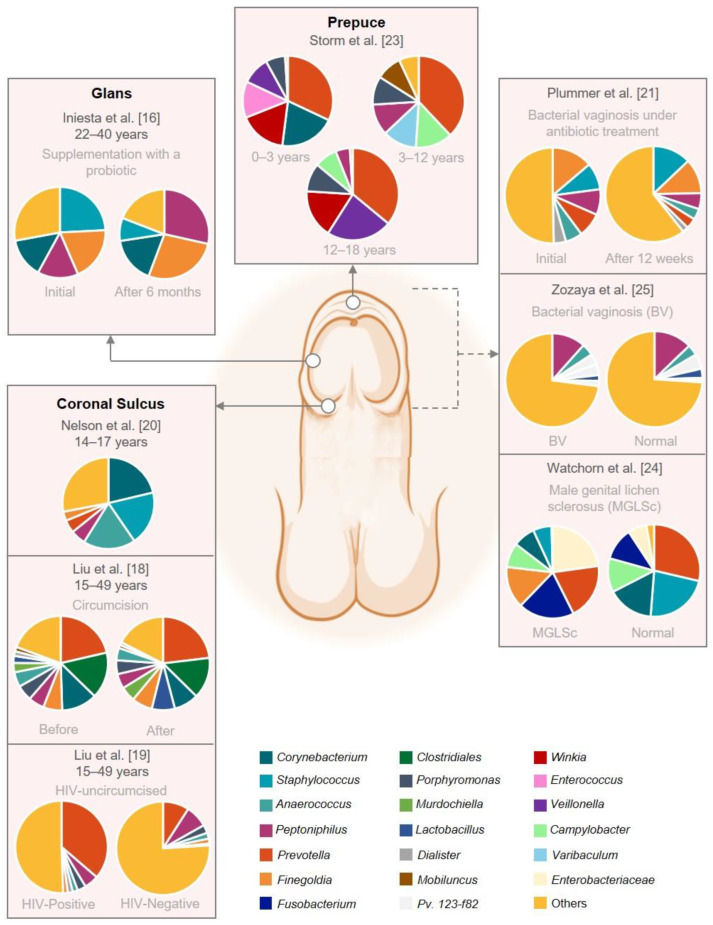
Graphical representation of the male genital mucosa microbiome composition. This figure was based on eight studies that reported the abundance on genus level. The remaining studies did not have raw abundance data available. The figure was created with BioRender.com (accessed on 16 September 2022).

**Table 1 microorganisms-10-02312-t001:** Overview of the final selected articles and the characteristics and reported findings. HV = healthy volunteers; CS = coronal sulcus; BP = balanoposthitis; HIV = human immunodeficiency virus; MGLSc = male genital lichen sclerosus; BS = balanopreputial sac; BV = Bacterial vaginosis; ↓ = decrease; ↑ = increase.

Author	Study Design	Field of Study	Participants	Age (Years)	Ethnicity, Country	Anatomical Site	Sample Type	Microbial Analysis	Key Findings	Limitations	Quality Score
Iniesta et al. [16]	Control intervention	Supplementation with a probiotic	17 couples under artificial reproductive treatment	Couples: 22–40 Male (mean 36) Female (mean 35)	Caucasian, Spain	Glans	Glans, semen, and blood	16S rRNA amplification of the V3–V4 region	Treatment with *Lactobacillus salivarius* PS11610 modified the microbiota composition improving the urogenital tract microbiome, solving the dysbiosis in 88.9% of the couples. Male samples showed higher bacterial diversity at genus level than female samples. Most prevalent genera in glans: Initial: *Peptoniphilus*, *Staphylococcus*, *Fineglodia*, and *Corynebacterium*; 3 and 6 months: *Peptonophilus*, *Finegoldia*, *Corynebacterium* (↓ *Staphylococcus*)	Small sample size. No control group treated with placebo.	Fair
Li et al. [17]	Case control	Balanoposthitis (BP)	26 BP uncircumcised 29 HV uncircumcised	18–65	Unknown, China	Glans, penis, and prepuce	Swabs	16S rRNA amplification of the V4 region	Microbiome BP ∼ HV, but ≠ HV with redundant prepuce. Most prevalent species BP: *Staphylococcus warneri* (with condom use) and *Prevotella bivia* (without sexual activity). Most prevalent HV: *Ezakiella* (redundant prepuce), *Porphyromonas somerae* (normal prepuce).	Small sample size. No ethnicity records. V4 region is considered a relatively low informative region for taxonomic assignment.	Fair
Liu et al. [18]	Randomized controlled trial	Circumcision	77 HV uncircumcised 79 HIV-negative pre- and post-circumcision	15–49	Unknown, USA	Coronal sulcus (CS)	Swabs	16S rRNA amplification of the V3-V6 region	Male circumcision reduced the prevalence and the absolute abundance of CS bacteria and the diversity of microbiota. Day 0: Prevalent but low abundance: *Prevotella* sp., *Clostridiales* and *Corynebacterium* sp. At 5%: *Peptoniphilus* sp., *Anaerococcus* sp., *Fenigoldia* sp., *Murdochiella* sp., *Porphyromonas* sp., and *Lactobacillus* sp. Year 1: Reduction in bacterial load on post-circumcision: *Porphymonas* sp., *Prevotella* sp., *Negativicoccus* sp., *Dialister* sp., *Mobiluncus* sp., and 6 genera from *Clostridiales* family XI. No reduction: *Atopolium* sp., *Sneathia* sp., and *Megasphaera*. More prevalent: *Kocuria* sp. and *Facklamia* sp.	No ethnicity records.	Good
Liu et al. [19]	Case control	Circumcision and Human immunodeficiency virus (HIV)	46 HIV-positive uncircumcised 136 HIV-negative uncircumcised	15–49	Unknown, Uganda	Coronal sulcus	Swabs	16S rRNA amplification of the V3–V4 region	HIV-positive uncircumcised: ↑ penile anaerobes. Most prevalent genera HIV-positive uncircumcised: *Prevotella*, *Dialister*, *Mobiluncus*, *Murdochiella*, and *Peptostreptococcus*.	No ethnicity records.	Good
Nelson et al. [20]	Observational cohort	Circumcision	18 HV (6 uncircumcised and 12 circumcised)	14–17	Mixed Black, Caucasian, and Latin, USA	Coronal sulcus	Swabs and first catch urine	16S rRNA amplification of the V1–V3, V3–V5, and V6–V9 regions	Microbiome CS ≠ urine. Most prevalent genera CS: *Corynebacteria*, *Staphylococcus*, *Anaerococcus*, *Peptoniphilus*, *Prevotella*, *Finegoldia*, *Porphyromonas*, *Propionibacterium*, and *Delftia*. Uncircumcised: ↑ *Prevotella* and *Porphyromonas*. Most prevalent genera urine: *Streptococcus*, *Lactobacillus*, *Gardnerella*, and *Veillonella*.	Small sample size.	Fair
Plummer et al. [21]	Randomized controlled trial	Bacterial vaginosis under antibiotic treatment	27 couples	>18	Unknown, Australia	Penis	Swab and first catch urine	16S rRNA amplification of the V3–V4 region	Day 0: Male specimens were heterogeneous in composition. Most abundant in penile swab: *Corynebacterium*, *Staphylococcus*, *Peptoniphilus*, and *Prevotella*. Day 8: Decreased on penile swab taxa—*Anaerococcus*, *Finegoldia*, *Peptoniphilus*, *Prevotela* spp., and *Dialister*. (↑ *Staphyloccocus*)	Small sample size. Self-collected swab.	Fair
Price et al. [22]	Randomized controlled trial	Circumcision	12 HIV-negative pre- and post-circumcision	15–49	Unknown, Uganda	Coronal sulcus	Swabs	16S rRNA amplification of the V4 region	Microbiome post-circumcision: ↓ penile anaerobes. Most prevalent genera pre-circumcision: *Anaerococcus*, *Peptoniphilus*, *Finegoldia*, and *Prevotella*. Most prevalent genera post-circumcision: *Staphylococcus* and *Corynebacterium*.	Small sample size. No ethnicity records.	Poor
Storm et al. [23]	Observational cohort	Healthy	48 HV males 18 HV females	0–18	Unknown, USA	Males: urethra, perineum, and foreskin Females: urethra, perineum, and vagina	Swabs	16S rRNA amplification of the V4 region	Perineal microbiomes differed significantly by age; urethral and foreskin microbiomes did not. Most common genera foreskin: *Prevotella*, *Staphylococcus*, *Corynebacterium*, *Peptoniphilus*, *Mobiluncus*, and *Winkia*.	Small sample size. No record of the pubertal status of the older cohort. No ethnicity records. V4 region is considered a relatively low informative region for taxonomic assignment.	Fair
Watchorn et al. [24]	Case control	Male genital lichen sclerosus (MGLSc)	20 MGLSc uncircumcised 20 HV uncircumcised	MGLSs:26–73 HV:19–63	Unknown, UK	Balanopreputial sac (BS) (Glans + inner prepuce)	Swabs and first catch urine	16S rRNA amplification of the V3–V4 region	Microbiome BS (MGLSc) ∼ urine (MGLSc). Microbiome BS (MGLSc) ≠ balanopreputial sac (HV). Most prevalent genera BS (MGLSc): *Enterobacteriaceae, Prevotella, Fusobacterium*, *and Finegoldia*. Most prevalent genera BS (HV): ↑ *Finegoldia*, *Staphyloccocus, Corynebacterium.*	Small sample size. No ethnicity records.	Poor
Zozaya et al. [25]	Cross-sectional	Bacterial vaginosis (BV)	65 HV-males (23 circumcised) 65 BV-males (35 circumcised)	Mean 30.7	African American, USA	Glans, coronal sulcus, penis	Swab	16S rRNA amplification of the V4-V6 region	More penile skin diversity of BV-males than normal-males, but urethral diversity did not differ between groups. BV-associated bacteria were more abundant in penile and urethral specimens on BV-males. Most abundant on penile skin of BV-males: *Peptoniphilus*, *Anaerococcus*, Pv. 123-f-82, Pv. 123-b-46, *Lactobacillus iners*, and Pv.123-f-110. Most abundant on penile skin of HV: *Peptoniphilus*, *Anaerococcus*, *Pv 123-f-82*, *L. iners*, *Porphyromonas*, and *Prevotela disiens*.	No records of recruitment duration.	Good

## Data Availability

Not applicable.

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
