# Peer review of "Microbiome in Male Genital Mucosa (Prepuce, Glans, and Coronal Sulcus): A Systematic Review"

_microorganisms, 2022, doi:10.3390/microorganisms10122312_

Round 1

Reviewer 1 Report

Studying the microbiome in male genital mucosa is essential for understanding infection risk and disease progression. The subject has an actual scientific interest, having significant implications for medical practice.

The topic is relevant and exciting to the field of the journal. The text is clear and easy to read. The manuscript has an excellent structure and description. The overall paper is organized and well-written. The literature reviews are insightful and informative. But, some references in the text need to fit with the title from the reference list. For example: on Line 218 is written Storm et al. [13] and at the reference at position 13 is Moher et al. and Storm et al. are on 23 places; on Line   308 is written Zozaya et al. [21], and Plummer et al. [21]. Please check all the references’ citations carefully again.

 The figures are well-presented and easy to read and understand. The presented aspects sufficiently support the conclusions. There is nothing about the limitations of the review.

I congratulate all the authors for their efforts.

Author Response

Thank you for the comments, the effort and the time taken to review the manuscript.

Thank you for pointing out the strengths to this study, including the clarity and straightforward description of the results which support the conclusions. As suggested by Reviewer, all the references’ citations were carefully checked. We changed reference in L128 (Storm et al.) and L316 (Zozaya et al.).

Reviewer 2 Report

This review considers an overview of microbiota in the male reproductive system. Therefore, the aim of this systematic review is to provide an overview of the microbial communities of healthy male genital mucosa and its role in disease. The search was limited to ten articles. The authors reported that the male genital microbiome composition shows similarities with the adjacent anatomical sites and is related to sexual intercourse. The limited data and paucity of prospective controlled studies highlight the need for additional studies and to establish criteria in sampling methods and in the microbiome assay procedures. This scenario is very interesting for the public journal. The articles included in this review are deficient. The discussion part is insufficient, the results of the different studies were not sufficiently compared. In the last, the conclusions should be improved, adding that studies are needed to improve the knowledgement in this field and introduce these exame in the clinical diagnostic field. In conclusion, check lines 165-167, improve the concept, and add tables that simplify the articles' results.

Author Response

Thank you for the comments, the effort and the time taken to review the manuscript. The ten articles included in this review meet the inclusion criteria that have been outlined and were carefully analyzed. However, there is still limited data on the microbiome of the male reproductive system, as we highlighted. For this reason, this systematic review does not include meta-analysis. Nevertheless, the studies included are congruent and we compare them with each other. For example, the studies that comprise circumcision and coronal sulcus microbiome (Nelson et al. 2012, Price et al. 2010, Liu et al. 2013, 2017) concluded all that circumcision alters the coronal sulcus microbiome, and anaerobic and vaginosis associated bacteria are the most abundant before circumcision, due to the existence of the moist anoxic microenvironment of the subpreputial space. These results are discussed in L307-L322.

As suggested by Reviewer we improved the discussion section. We improved L295-298, regarding the microbiome of prepuce and L345-L350 about the limitations of the current literature. The comparison of the number of participants, study setting, and study methodology have already been compared (L337-L383).

As suggested by Reviewer we also improved the conclusions section. We added the studies which are necessary to improve the knowledge in this field, such as pyrosequencing, whole metagenome sequencing and quantitative real-time PCR, and how this is applied in clinical practice (e.g., potential biomarkers for new therapeutic and prognostic options). We also clarified the L167-168 as suggested.

We believe that these suggestions improved the manuscript.

Reviewer 3 Report

This manuscript covers interesting and emerging topic of microbiome in male genital mucosa. Even though similar studies were and are constantly performed, the topic itself is still actual and demands every day improvement.

Minor corrections and editions throughout the text together with nicely descibed discussion could throttle future research and publications on the described topic.

Discussion: while topic of microbiota changes throughout different studies was widelly covered, it would be beneficial for readers to briefly describe methods by which microbiota is studied; i.e., metaproteomics (doi.org/10.3390/microorganisms9050980), metagenomics (doi.org/10.3390/microorganisms10040711), trancriptomics (doi.org/10.3390/app12052483), etc.

Otherwise, I would like to greet authors with a nicely written manuscript and wish them further success.

Author Response

Thank you for the comments, the effort and the time taken to review the manuscript.

Thanks for enjoying the discussion. As suggested by Reviewer, methods by which microbiota can be studied were added (L263-270).

Round 2

Reviewer 2 Report

This review was improved and the suggestions were made.